# Graph Theory for Modeling and Analysis of the Human Lymphatic System

**Rostislav Savinkov** [1,2,3,*,†], **Dmitry Grebennikov** [1,2,4,*,†], **Darya Puchkova** [5],
**Valery Chereshnev** [6], **Igor Sazonov** [7] **and Gennady Bocharov** [1,2,4,*]

1  Marchuk Institute of Numerical Mathematics, Russian Academy of Sciences (INM RAS),
   119333 Moscow, Russia
2  Moscow Center for Fundamental and Applied Mathematics at INM RAS, 119333 Moscow, Russia
3  Nikolsky Mathematical Institute, Peoples' Friendship University of Russia (RUDN University),
   117198 Moscow, Russia
4  Institute for Personalized Medicine, Sechenov First Moscow State Medical University,
   119991 Moscow, Russia
5  Moscow Institute of Physics and Technology, National Research University,
   141700 Dolgoprudny, Moscow Region, Russia; darya.puchkova@phystech.edu
6  Institute of Immunology and Physiology, Ural Branch of Russian Academy of Sciences,
   620000 Yekaterinburg, Russia; mchereshneva@mail.ru
7  College of Engineering, Swansea University, Swansea SA1 8EN, UK; i.sazonov@swansea.ac.uk
*  Correspondence: r.savinkov@inm.ras.ru (R.S.); dmitry.ew@gmail.com (D.G.); g.bocharov@inm.ras.ru (G.B.)
†  These authors contributed equally to this work.

**Abstract:** The human lymphatic system (HLS) is a complex network of lymphatic organs linked through the lymphatic vessels. We present a graph theory-based approach to model and analyze the human lymphatic network. Two different methods of building a graph are considered: the method using anatomical data directly and the method based on a system of rules derived from structural analysis of HLS. A simple anatomical data-based graph is converted to an oriented graph by quantifying the steady-state fluid balance in the lymphatic network with the use of the Poiseuille equation in vessels and the mass conservation at vessel junctions. A computational algorithm for the generation of the rule-based random graph is developed and implemented. Some fundamental characteristics of the two types of HLS graph models are analyzed using different metrics such as graph energy, clustering, robustness, etc.

**Keywords:** graph theory; networks; mathematical modeling; lymphatic system; immunology

## 1. Introduction

The human lymphatic system (HLS) is a complex network of lymphatic organs linked through the lymphatic vessels. Its structure and functioning are critical to immunity by transporting the immune cells and antigens [1] and maintaining the fluid balance in tissues [2]. Currently, there are only a few studies considering the spatial structure of the human lymphatic system in terms of the network models [3–5] . The mathematical properties of the HLS network remain poorly investigated. The graph theory provides a powerful tool to implement, visualize and analyze the characteristics of the network models. Recently, it has been successfully applied for analysis of the topological properties and robustness of conduit networks in lymphatic nodes [6].

Generally, two complementary approaches to modeling the network structure of the lymphatic system. One is based on employment of available anatomical data [7,8]. The second approach uses some general rules of lymphatic vessels network organization as discussed in [5,9]. In this study,

we develop oriented graph-type models of the HLS following both approaches. To specify the links direction in the anatomically derived HLS graph model, the analysis of the homeostatic lymph flow through the network is performed using the Poiseuille law. The topological characteristics of the graph models are analyzed using some general metrics such as graph energy, clustering, robustness, etc. The aim of the present study is to explore the fundamental mathematical properties of the lymphatic vessels network, for which the guiding organizational principles remain to be identified [2].

The paper is structured as follows. In Section 2, we present two anatomical data-based graph models, i.e., the Reddy's [3] and the Plastic boy model [5] of the HLS. The lymph flow balance is analyzed to specify the respective directed graph of the HLS. In Section 3, computational algorithm for generating a random rule-based directed graph of the HLS is formulated and applied to derive the rule-based model. In Section 4, the parameters of the algorithm for the rule-based graph model are estimated to best-fit the topological characteristics of the anatomy-based model. In Section 5, a systematics analysis of the anatomy-based and the rule-based graph models of the HLS is performed. Conclusions and future work are discussed in the last section.

## 2. Graph Models Based on Anatomical Data

The physiological data on the structural organization of the HLS provide a foundation for developing anatomically based models of the HLS network. These days, the input information can be obtained from the Plasticboy project (http://www.plasticboy.co.uk/store/Human_Lymphatic_System_no_textures.html) or the CGTrader (2011–2020) at https://www.cgtrader.com/3d-models/character/anatomy/lymphatic-system-inhuman-body.

### 2.1. Reddy's Directed Graph Model of the HLS

The first network model of the HLS has been systematically developed by Reddy [3]. Due to the scarcity of the available data at that time, the network model considers only the major lymphatic vessels, many organs are lumped into single vessels and the right side of the head, neck, thorax and upper right extremity are nor considered. The simplified Reddy's network model of the HLS can be represented by oriented graph $G(V, E)$ with 29 vertices and 28 edges as shown in Figure 1 generated using the network analysis package *igraph* (https://igraph.org/).

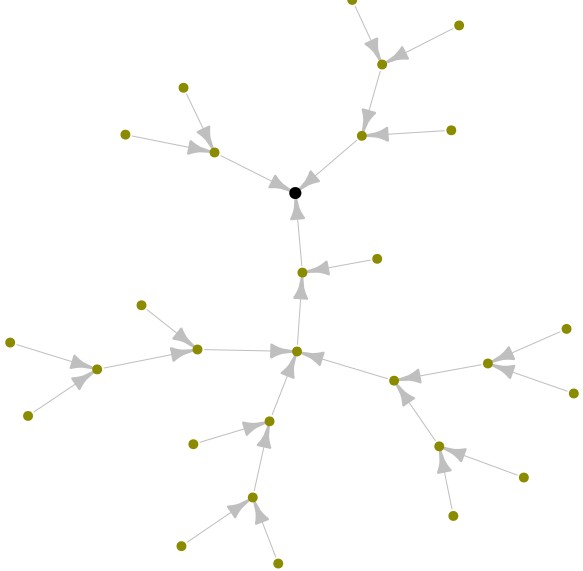

**Figure 1.** Oriented graph of a simple network model of the human lymphatic system suggested by Reddy et al. [3] with 29 nodes and 28 edges. The black node with out-degree $deg^+ = 0$ corresponds to jugular vein (sink).

### 2.2. Plastic Boy-Derived Graph Model of the HLS

We have previously developed a 3D computational geometry model of the HLS based on available anatomical data from the *Plasticboy* project [5]. This network is a graph $G(V, E)$ containing 996 vertices and 1117 edges as shown in Figure 2. Unlike the Reddy's graph model of the HLS shown in the Figure 1, the above model is an undirected graph. To transform it to an oriented graph, we need to determine the directions of the lymph flow through the graph. To this end, the simulation of a steady lymph flow based on Poiseuille equation is employed.

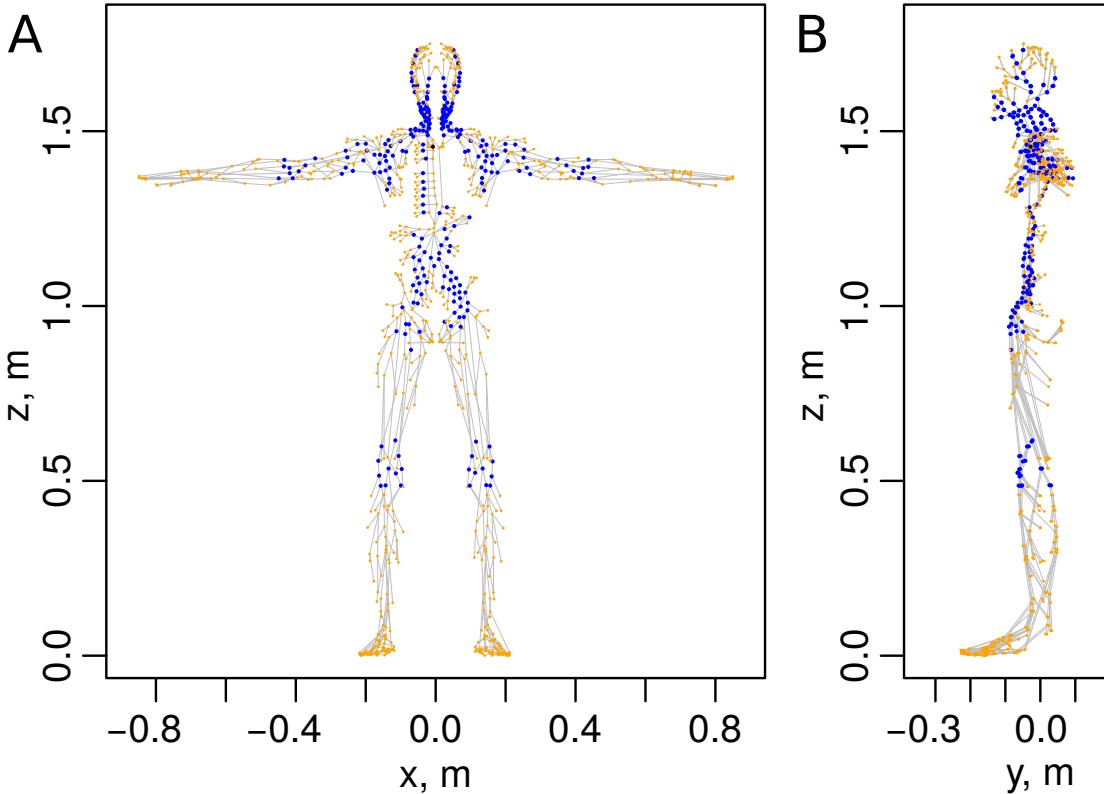

**Figure 2.** Graph of a human lymphatic system based on anatomical *Plasticboy* data [5] with 996 vertices and 1117 edges (graph files are provided in Supplementary Materials). (**A**) Face- and (**B**) side views. Blue colored vertices represent 272 lymph nodes. The black vertex corresponds to the output node.

### 2.2.1. Undirected Graph of the HLS

The anatomical data-based 3D graph model contains several vertices of degree 2 corresponding to the lymphatic vessels which are composed of several lymphangions. We remove from the graph all vertices of degree 2 corresponding to individual lymphangions to represent the multi-lymphangion vessels as single graph edges. Moreover, we add one edge to correct the connection of right arm to the body (which was missing in the original graph) and three edges to get a single output vertex for the whole HLS graph (modeling the excretory vessels emptying the lymph to the venous system, i.e., the jugular lymphatic trunk, thoracic duct and jugular vein). Overall, the following specifications are made to obtain the HLS presented in Figure 2:

1. coordinates of all vertices in the graph are scaled to correspond to the basal 1750 mm height of a man;
2. the right hand and the head right side vertices are attached to the major part of HLS graph by adding the edge from vertex 987 to the vertex 993;

3.  single output node (sink): (1) two vertices are added (numbered as 995 and 996) with vertex 996 being the output of the system, (2) three edges, i.e., between nodes 993–995, 994–995 and 995–996 were added;
4.  we iterate over graph vertices and search for irrelevant vertices that meet all the following conditions:

    (a)  their degree is equal to two;
    (b)  they are not lymph nodes;
    (c)  their neighbor vertices $v_a$, $v_b$ are not connected with each other by an edge;

    at each iteration, the irrelevant vertex was removed, and the edge $v_a - v_b$ was created.

The topological properties of the HLS data-based graph models, i.e., the vertices degree- and edge-length distributions for the Plastic boy data-based graph and Reddy's graph of the HLS are summarized in Figure 3A,B, respectively. For the data-based HLS graph, we additionally show the edge-length distribution and degree distribution of the vertices that represent lymph nodes (LNs) in Figure 3A.

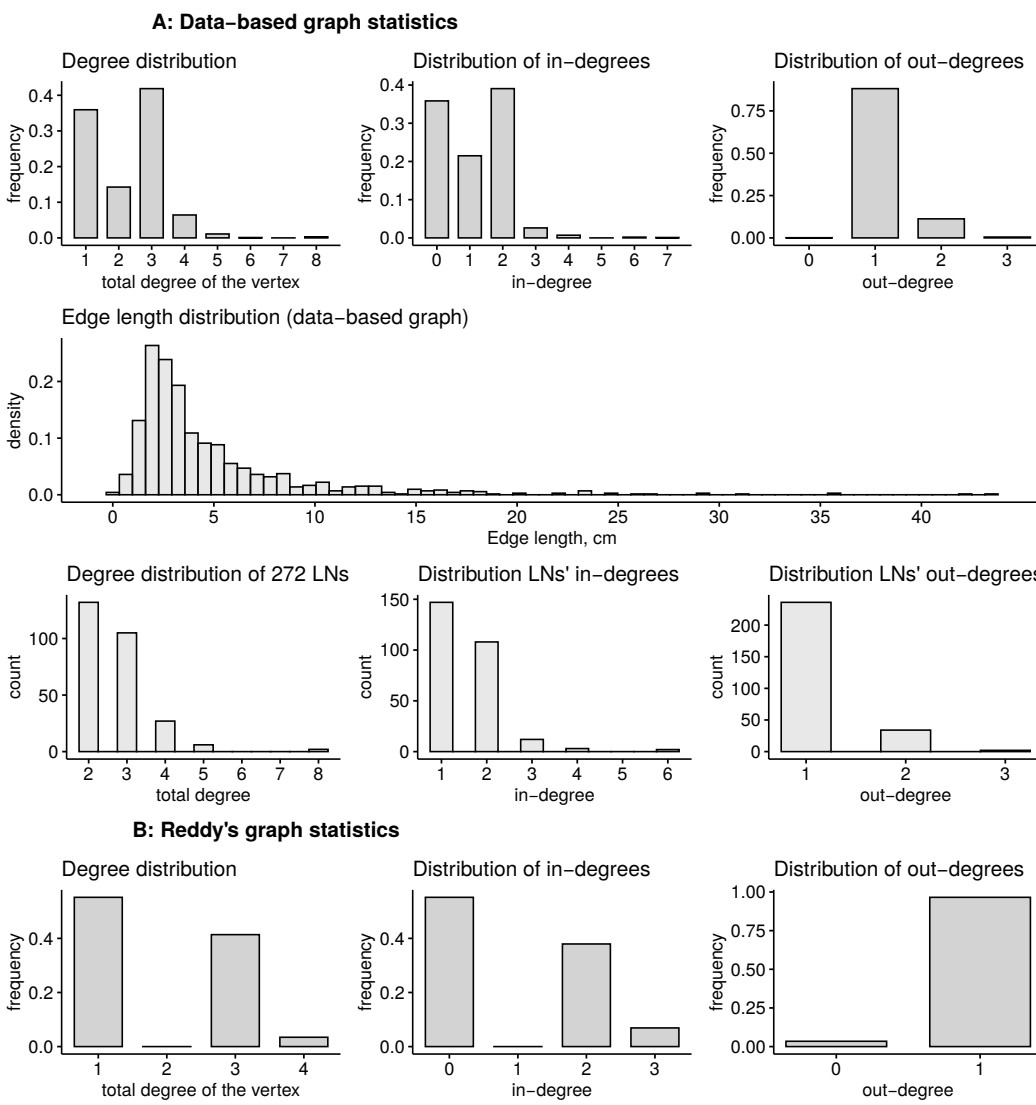

**Figure 3.** Statistical distributions of the anatomy data-based HLS graphs. (**A**) Statistics of the data-based graph, (**B**) of the Reddy's graph. Statistics include the distributions of total degrees of the vertices, distributions of in- and out-degrees, as well as the edge-length distribution and degree distribution of the vertices that represent the lymph nodes (LNs) for data-based graph.

### 2.2.2. Lymph Flow Analysis for Specifying the Directed Graph

The lymphatic system collects excess of intercellular fluid in the body tissues, drains it through the lymph nodes (LN) and redirects the lymph further along the lymphatic vessels, delivering the collected fluid into the central venous duct. In the LNs, about 90% of the afferent lymph flow takes a peripheral path through the subcapsular and medullary sinuses [10]. The remaining 10% enters interstitium to finally reach the efferent lymphatic vessel or to be absorbed by parenchymal blood vessels of the LN. Earlier experimental data show that about 10% of the lymph is absorbed into the blood vessels that penetrate the lymph nodes [11,12], while the remaining 90% of lymph entering LN is transported further through the lymphatic vessels to higher-level LN.

We study the balance of lymph flow in the undirected anatomical-based graph presented in Figure 2 to obtain the direction of lymph flow through the graph. We use the similar approach as described in [13]. We apply the Poiseuille equation to every internal vessel represented by edge $(i, j)$,

$$Q_{ij} = \frac{\pi R_{ij}^4 (p_i - p_j)}{8 \eta L_{ij}} \tag{1}$$

where $Q_{ij}$ is the lymph flow rate along the vessel, $R_{ij} = R$ is the vessel lumen radius, $\eta$ is the dynamic viscosity of the lymph, $p_i$, $p_j$ are the pressures at the ends of the vessel, $L_{ij}$ is the length of the vessel. In the internal nodes, the mass conservation law is imposed:

$$\sum_j Q_{ij} = 0, \quad i = 1, ..., N - N_{inp} - N_{out}, \tag{2}$$

where $N_{inp} = 357$ is the number of input nodes with zero in-degree that provide the inflow of the lymph $Q_{in} = 0.081 \text{ mm}^3/\text{s}$, $N_{out} = 1$ is the only output node with outflow $Q_{out} = N_{inp} Q_{in} = 28.93 \text{ mm}^3/\text{s}$. To close the system (1) and (2) we specify the pressure $p_{out} = 1127.73$ Pa on the output node and $Q_{in}$ flow rate on the input nodes.

The above set of equations on HLS network in the matrix form

$$\mathbf{Kp} = \mathbf{b}, \tag{3}$$

to calculate the lymph flows where $\mathbf{p} = [p_1, \ldots, p_N]^T$ is an unknown pressure vector for all the vertices of the graph $G(V, E)$. Matrix $\mathbf{K}$ and vector $\mathbf{b}$ are defined as

$$K_{ij} = \begin{cases} -A_{ij} \frac{\pi R^4}{8 \eta L_{ij}} & j \neq i \quad i \in V_{inp} \cup V_{int} \\ \sum_{k=1}^{N} A_{ik} \frac{\pi R^4}{8 \eta L_{ik}} & j = i \quad i \in V_{inp} \cup V_{int} \\ \delta_{ij} & i \in V_{out} \end{cases} \qquad b_i = \begin{cases} Q_{in} & i \in V_{inp} \\ 0 & i \in V_{int} \\ P_{out} & i \in V_{out} \end{cases} \tag{4}$$

where $A$ is an adjacency matrix of the LS graph; $L_{ij}$ is a distance between vertices $i$ and $j$; $V_{inp}, V_{int}, V_{out}$ are sets of input, internal and output vertices; $\delta_{ij}$ is Kronecker's delta. For the inflow vertices, we assume that all input vertices obtain equal amount of lymph ($Q_{in}$). For internal vertices, the conservation of mass requires the sum of incoming and outgoing flows to be zero. For the outflow vertices, we fix the pressure values $P_{out}$.

To calculate the stationary flows, the following assumptions were made:

1. there are no other sinks of the lymph except for the exit vertex number 996: $V_{out} = \{996\}$;
2. all vertices of the graph that have only one connection with other vertices (except for the vertex 996) are the points of lymph entry (collecting lymphatics) into the lymphatic network;
3. the pressure at all inflow vertices is adjusted to provide the lymph flow rate on the input edges to be equal to the output flow rate of the system divided by number of vertices with zero input edges;
4. the radii of all the vessels are set to be same, equal to 1 mm;
5. the constant viscosity is set in the lymphatic network.

The value of the dynamic viscosity of lymph at a temperature of 37 °C ($\eta = 1.8 \times 10^{-3}$ Pa·s) is used. The pressure in the central venous duct ranges from 8 to 15 mmHg [2,14–16], and the value of 11.5 mmHg is considered in the calculations. According to existing data, from 2 to 3 L of lymph per day gets into the central venous duct [17,18], hence we take this parameter to be 2.5 L per day. The resulting stationary lymph flow distribution in the HLS is shown in Figure 4.

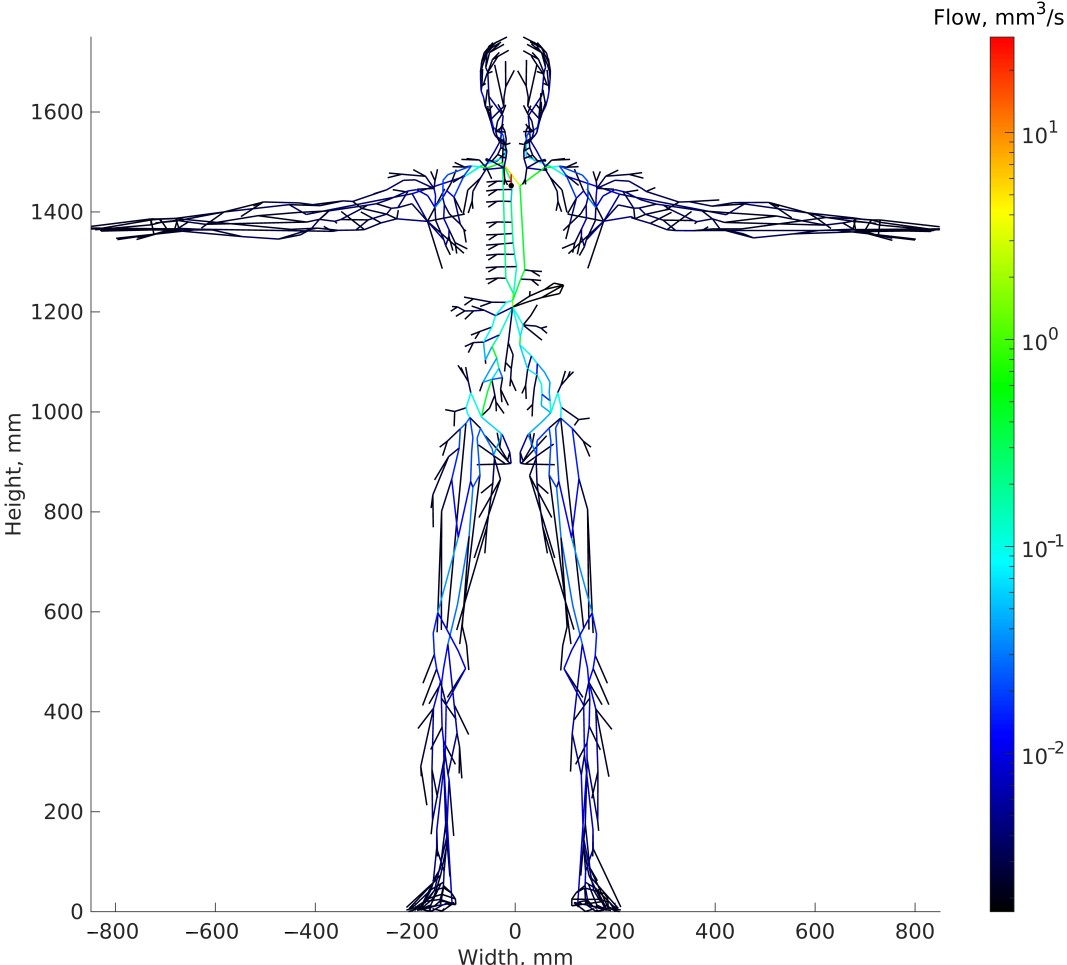

**Figure 4.** Steady-statedistribution of lymph flows in the HLS graph model. The output sink node is presented as black circle.

The above analysis enabled us to specify the lymph flow directions in the graph edges, and ultimately, to generate an oriented graph of the human lymphatic system, which is shown in Figure 5A. As the dimension of the graph HLS model is high, it can be visualized in 2D being subdivided into six segments representing specific parts of the human organism: arms, legs, head and body. The resulting oriented subgraphs are presented in the Figure 5B.

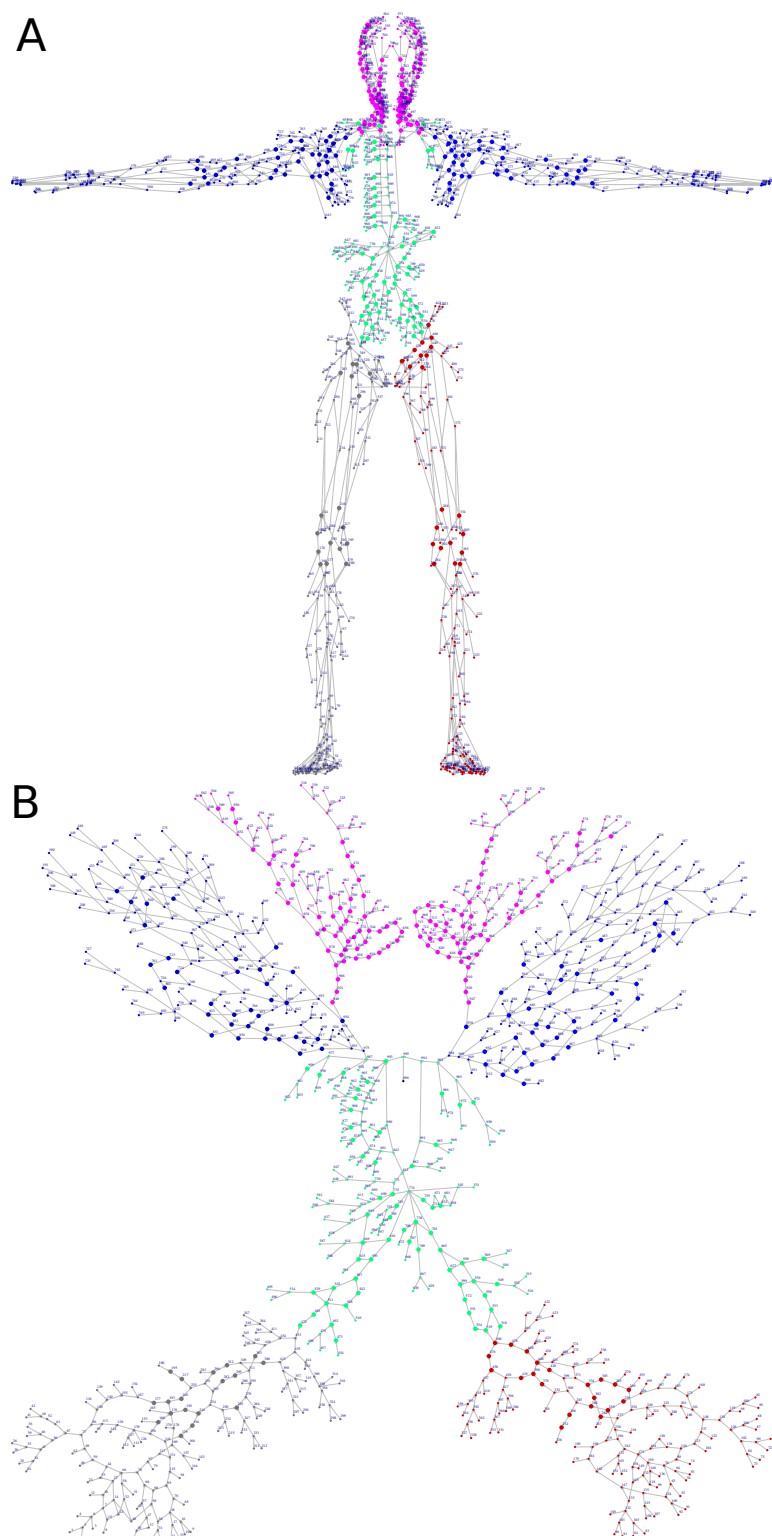

**Figure 5.** Oriented graph model of the HLS based on the analysis of the lymph flow directions in the network. (**A**) Spatial view of the graph with numbered vertices (face-view projection of 3D coordinates). (**B**) View of the graph on a plane with no-overlapping layout. The 2D layout was obtained using spring-based model algorithm [19] implemented in package *igraph*, followed by some manual tweaking to space out the vertices. The colored subgraphs correspond to different parts of the body, i.e., arms (dark and light blue), head/neck (magenta), torso (light green), legs (red and gray). LNs are represented by larger circles.

The adjacency matrix of this graph is presented in Figure 6.

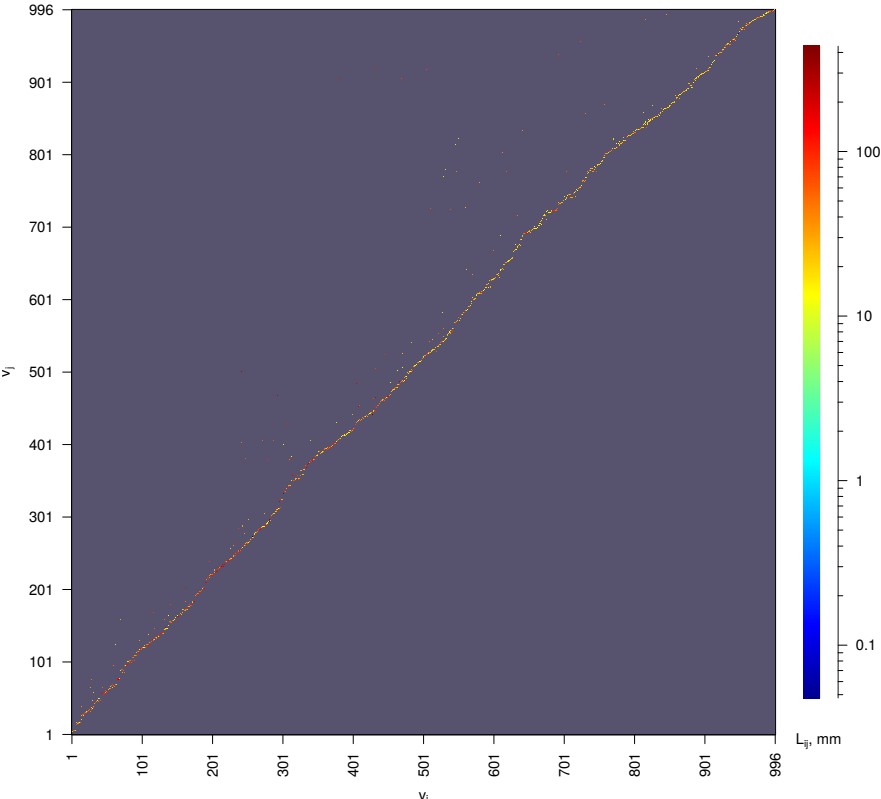

**Figure 6.** Adjacency matrix of the oriented data-based graph of human lymphatic system shown in Figure 5. Colors represent the lengths $L_{ij}$ of the edges $e_{ij}$.

## 3. Computational Algorithm for Generating a Random Rule-Based Directed Graph of the HLS

A quantitative modeling of the HLS presents a challenge due to enormous morphological complexity and variability in its appearance [5]. Scarcity of available anatomical data calls for the development of the so-called rule-based models of the HLS network structure. General rules of the lymphatic vessels network organization can be specified as follows: (*i*) no long-distance edges are allowed, (*ii*) nodes can have multiple inflow connections, (*iii*) the nodes connections are locally acyclic, (*iv*) the lymphatic system network is a circular graph [9], and additionally, the links between nodes of the same layer are allowed. To build a random graph of the lymphatic system, we developed a rule-based algorithm that generates a random oriented graph with a predetermined number of layers $N_l$, number of vertices $N_v$, number of sources $N_{inp}$ (input nodes, which should only have out-degree 1) and with one sink (output node). The layers in the algorithm are numerated from the output to the input vertices, from *ground layer* with index 1 to *top layer* with index $N_l$. Iterative process on vertices is executed layer by layer starting from the next to the top layer. The rule-based graph HLS model generation consists of three major steps as specified in Algorithm 1 (C++ language realization of the algorithm is provided in Supplementary Materials, for any questions please contact authors of the article).

---

**Algorithm 1:** Generation of a rule-based directed graph of the HLS.

---

**Parameters:** $N_v$, $N_{inp}$, $N_l$, $P_e$, $P_o$;

$\quad\quad\quad\quad$ $N_v$—number of vertices; $N_{inp}$—number of input nodes; $N_l$—number of layers;

$\quad\quad\quad\quad$ $P_e$—probability of additional edge creation; $P_o$—prob. of over-layer edge type;

Init array $A$ of size $N_l$ to store lists of vertices in each layer;

---

**Step 1.** Distribute the vertices through the layers.

Set $AVL = round(1.1 \cdot (N_v - 1)/(N_l - 1))$;

**for** *each layer L from A* **do**

$\quad\quad$| add one vertex to the layer $L$;

**end**

**while** *vertices count in A is less than $N_v$* **do**

$\quad\quad$ **select** random layer $L$ from A;

$\quad\quad$ **if** *L is a top layer of A **and** vertices count in layer L is less than **min**$(N_{inp},\ AVL)$* **then**

$\quad\quad\quad\quad$| add one vertex to the layer $L$;

$\quad\quad$ **else**

$\quad\quad\quad\quad$ **if** *L is **not** top layer of A **and** vertices count in layer $(L + 1)$ is more than in layer L* **then**

$\quad\quad\quad\quad\quad\quad$| add one vertex to the layer $L$;

$\quad\quad\quad\quad$ **end**

$\quad\quad$ **end**

**end**

---

**Step 2.** Configure the input vertices.

Init empty list $L_{inp}$ to store the input vertices (which can be spread through the layers);

insert all vertices from *top layer* of A into $L_{inp}$;

**while** *vertices count in $L_{inp}$ is less than $N_{inp}$* **do**

$\quad\quad$ **select** random layer $L_c$ from A;

$\quad\quad$ let $n_c$ be the number of vertices in layer $L_c$;

$\quad\quad$ let $n_{inp}$ be the number of vertices in layer $L_c$ that are present in $L_{inp}$;

$\quad\quad$ **if** $n_c > n_{inp} + 1$ **then**

$\quad\quad\quad\quad$ **select** random vertex from layer $L_c$ that is **not** in $L_{inp}$ as $v_{candidate}$;

$\quad\quad\quad\quad$ insert $v_{candidate}$ into $L_{inp}$;

$\quad\quad$ **end**

**end**

---

**Step 3.** Create directed edges between the vertices.

**for** *each vertex v in A* **do**

$\quad\quad$ let $L_v$ be the layer which holds the current vertex $v$;

$\quad\quad$ let $L_{first}$ be the list of vertices from layer $(L_v - 1)$ that do not have inflow connections;

$\quad\quad$ let $L_{second}$ be the list of vertices from layer $(L_v - 1)$ that do have inflow connections;

$\quad\quad$ $(\deg^-(v) = 0$ for $v \in L_{first},\ \deg^-(v) > 0$ for $v \in L_{second})$

$\quad\quad$ **if** $L_{first}$ *is **not** empty* **then**

$\quad\quad\quad\quad$| **select** random vertex from $L_{first}$ that is **not** in $L_{inp}$ as $v_s$;

$\quad\quad$ **else**

$\quad\quad\quad\quad$| **select** random vertex from $L_{second}$ that is **not** in $L_{inp}$ as $v_s$;

$\quad\quad$ **end**

$\quad\quad$ create directed edge from current vertex $v$ to selected vertex $v_s$;

$\quad\quad$ **if** *v is **not** in $L_{inp}$* **then**

$\quad\quad\quad\quad$ **while** $P_e > random(0,1)$ **do**

$\quad\quad\quad\quad\quad\quad$ let $L_{tv} = L_v - 1$;

$\quad\quad\quad\quad\quad\quad$ **while** $L_{tv} > 1$ **and** $P_o > random(0,1)$ **do**

$\quad\quad\quad\quad\quad\quad\quad\quad$| $L_{tv} = L_{tv} - 1$;

$\quad\quad\quad\quad\quad\quad$ **end**

$\quad\quad\quad\quad\quad\quad$ **select** random vertex from layer $L_{tv}$ that is **not** in $L_{inp}$ as $v_s$;

$\quad\quad\quad\quad\quad\quad$ create directed edge from current vertex $v$ to selected vertex $v_s$;

$\quad\quad\quad\quad$ **end**

$\quad\quad$ **end**

**end**

---

The optimal estimation of the algorithm parameters is performed in Section 4. To this end a topological fitness function specifying a mismatch between the anatomy-based graph and the rule-based graph is used. One realization of a random graph constructed using Algorithm 1 with the optimal parameters is presented in Figure 7, its degree distribution is shown in Figure 8, its adjacency matrix—in Figure 9.

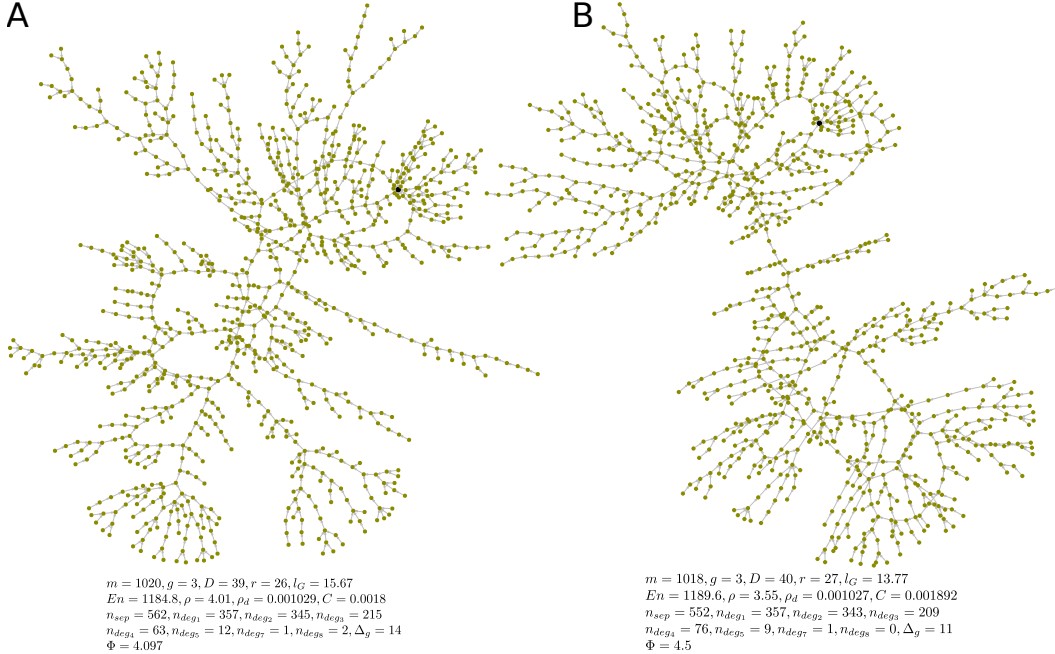

$m = 1020, g = 3, D = 39, r = 26, l_G = 15.67$
$En = 1184.8, \rho = 4.01, \rho_d = 0.001029, C = 0.0018$
$n_{sep} = 562, n_{deg_1} = 357, n_{deg_2} = 345, n_{deg_3} = 215$
$n_{deg_4} = 63, n_{deg_5} = 12, n_{deg_7} = 1, n_{deg_8} = 2, \Delta_g = 14$
$\Phi = 4.097$

$m = 1018, g = 3, D = 40, r = 27, l_G = 13.77$
$En = 1189.6, \rho = 3.55, \rho_d = 0.001027, C = 0.001892$
$n_{sep} = 552, n_{deg_1} = 357, n_{deg_2} = 343, n_{deg_3} = 209$
$n_{deg_4} = 76, n_{deg_5} = 9, n_{deg_7} = 1, n_{deg_8} = 0, \Delta_g = 11$
$\Phi = 4.5$

**Figure 7.** Two examples (**A**,**B**) of algorithmically generated realizations of rule-based directed random graphs of the HLS. The output nodes are marked with black color. Algorithm parameters: $N_v = 996$, $N_{inp} = 357$, $N_l = 41$, $P_e = 0.035$, $P_o = 0.21$. The metrics of topological fitness to anatomy-based graphs presented in figure are defined in Section 4.

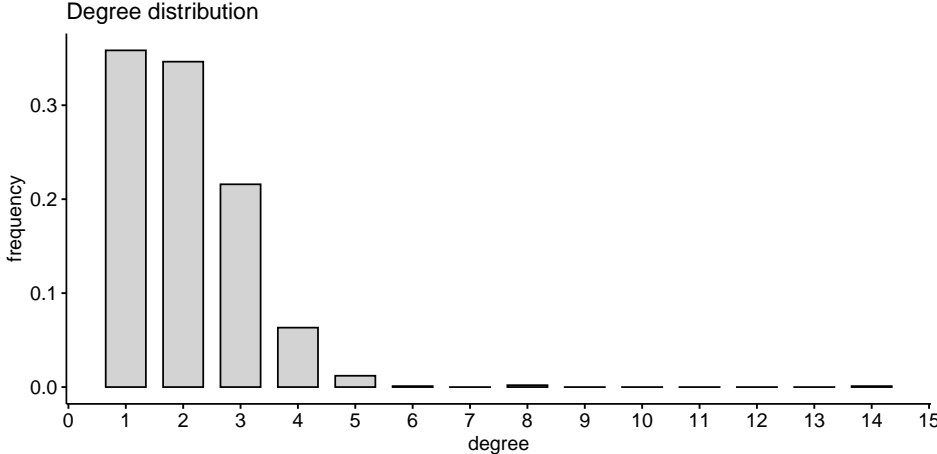

**Figure 8.** Degree distribution of the rule-based graph presented in Figure 7A.

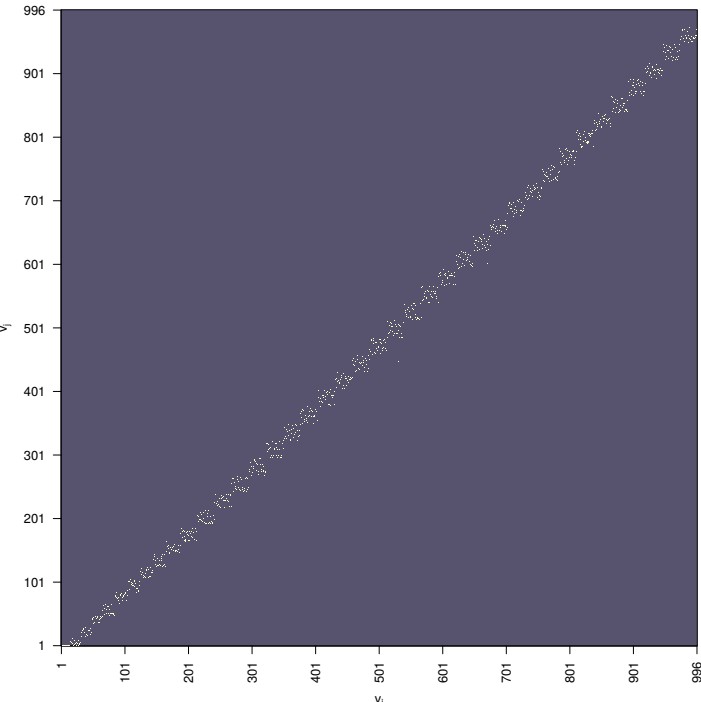

**Figure 9.** Directed graph adjacency matrix of the rule-based random graph presented in Figure 7A.

## 4. Parameters of the Rule-Based Algorithm Providing Best Match to the Anatomy Data Graph

The basic topological properties of the graph are the node (vertex) degree- and the edge (arc) length distributions. These were presented for two types of graph in the previous sections. As the coordinates of nodes (and edge lengths) of the rule-based graph are not specified, we can only obtain statistics on the degree distribution of the vertices, and this indicator varies significantly depending on the parameter settings of the algorithm. We consider some fundamental characteristics of the HLS graph properties to make a deeper comparison of the graph models such as the energy, clustering, robustness, etc. Let $G = (V, E)$ be an undirected version of the graph with $n = |V(G)|$ and $m = |E(G)|$ nodes and edges, respectively. Let $A$ denote the $n \times n$ adjacency matrix of $G$.

- The number of input nodes $N_{inp}$, i.e., the number of nodes with degree 1 and out-degree 0.
- Maximum degree of graph $\Delta_G$, i.e., the maximum degree of its vertices.
- Girth of the graph $g$, which is the length of the shortest (undirected) cycle in the graph.
- Diameter, i.e., the longest geodesic distance (in other terms, maximum eccentricity of any vertex),

$$D = \max_{v \in V} \epsilon(v) = \max_{v \in V} \max_{u \in V} d(u, v), \tag{5}$$

where $d(u, v)$ is the geodesic distance (shortest directed path connecting vertices $u$ and $v$), $\epsilon(v)$ is the eccentricity of vertex $v$.
- Radius of the graph (minimum eccentricity of any vertex),

$$r = \min_{v \in V} \epsilon(v) = \min_{v \in V} \max_{u \in V} d(u, v). \tag{6}$$

- Average path length (mean geodesic distance),

$$l_G = \frac{1}{n(n-1)} \sum_{u,v \in V, \, u \neq v} d(u, v). \tag{7}$$

- The energy and the spectral radius of the graph are defined as follows,

$$En(A) = \sum_{j=1}^{n} |\lambda_j|, \quad \rho(A) = \max\{|\lambda_j|\}, \tag{8}$$

where $\lambda_j$ stand for the eigenvalues of the adjacency matrix $A$ of the graph.

- Edge density of the graph, i.e., the number of edges divided by the number of all possible edges,

$$\rho_d = \frac{m}{n(n-1)}. \tag{9}$$

- The clustering coefficient $C$ (transitivity) measures the probability that two neighbors of a vertex are connected. It can be computed as function of adjacency matrix $A$:

$$C(A) = \frac{\sum_{i=1,j=1,k=1}^{n,n,n} a_{ij} \cdot a_{jk} \cdot a_{ki}}{\sum_{i=1}^{n} ((\sum_{j=1}^{n} a_{ij}) \cdot ((\sum_{j=1}^{n} a_{ij}) - 1))}. \tag{10}$$

- Number of separators $n_{sep}$, i.e., the vertices removal of which disconnects the graph.
- The robustness $R$ of the graph can be defined as the fraction of peripheral vertices that retained the connection with the output vertex after removing 5% of the vertices selected randomly, averaged over $k = 1, ..., N_r$ removal realizations. Given adjacency matrix $\tilde{A}_k$ of the $k$-th realization of the graph, indices $i_1, ..., i_{n_{inp}}$ of its input vertices and index $o$ of the output vertex, the robustness is computed as

$$R = \frac{1}{N_r} \sum_{k=1}^{N_r} R_k, \quad R_k = \sum_{i=1}^{n_{inp}} \frac{F(\tilde{A}_k, i_i, o, n)}{n_{inp}}, \quad F(\tilde{A}_k, i_i, o, n) = \begin{cases} 1, & if \ \sum_{j=1}^{n-1} |(\tilde{A}_k^j)_{i_i,o}| > 0, \\ 0, & if \ \sum_{j=1}^{n-1} |(\tilde{A}_k^j)_{i_i,o}| = 0, \end{cases} \tag{11}$$

- Topological diversity of the vertices as a function of the Shannon entropy associated with flow rates through the incident edges,

$$D_{flow}(v_i) = \frac{H(v_i)}{\log(k)} = \frac{-\sum_{j=1}^{k} p_{ij} \log(p_{ij})}{\log(k)}, \quad p_{ij} = \frac{|Q_{ij}|}{\sum_{j=1}^{k} |Q_{ij}|}, \tag{12}$$

where $k$ is the number of $v_i$'s incident edges and $p_{ij}$ is the proportion of the flow between the adjacent $v_i$ and $v_j$ to the total flow through the edges involving $v_i$. The flow diversity is defined analogous to the social and spatial diversity of networks from [20].

The rule-based graphs generated with Algorithm 1 have five tuning parameters: (1) number of vertices $N_v$, (2) number of input vertices $N_{inp}$, (3) number of layers $N_l$, (4) probability of new edge creation $P_e$ at each step of the algorithm, probability that the created edge connects vertices from different layers $P_o$. The first two parameters are set to match explicitly the properties of the anatomy-based graph: $N_v = 996$, $N_{inp} = 357$. The number of layers can be estimated as $N_l = D + 1 = 41$, where $D$ is the diameter of the directed target graph, which is increased by one because the output vertex is represented as a separate (ground) layer in Algorithm 1. Given the metrics characterizing the topology of an anatomy-based graph, we search for values of the remaining parameters $P_e$ and $P_o$ that can produce the graphs with similar topological structure. Specifically, we introduce the following state-vector

$$s(G) = \left( m, g, D, r, l_G, En, \rho, \rho_d, C, n_{sep}, n_{deg_1}, n_{deg_2}, n_{deg_3}, n_{deg_4}, n_{deg_5}, n_{deg_7}, n_{deg_8}, \Delta_G \right)^T, \tag{13}$$

which describes the topological properties of graph $G(V, E)$, and the objective function

$$\Phi(G) = \sum_{i=1}^{18} \left( \frac{s_i(G) - s_i(G^*)}{s_i(G^*)} \right)^2, \tag{14}$$

which penalizes the topological discrepancies of graph $G$ from the target graph $G^*$ and weighs them with $(s_i(G^*))^{-2}$ to bring discrepancies of different components of vector $s$ to a single scale. Here, we denote the number of vertices with degree $i$ as $n_{deg_i} = \sum_{v \in V,\, deg(v)=i} 1$, and exclude from the state-vector $n_{deg_6}$, which equals zero for an anatomy-based graph, to avoid singularity in (14).

First, we investigate the landscape of topological fitness of algorithmically generated graphs over uniform ranges of admissible values of parameters $P_e$ and $P_o$. The dependence of the objective function (14) on $P_e$ and $P_o$ is presented in Figure 10A. The colors correspond to logarithmically transformed mean values of objective function $\Phi_{mean}$ over 640 algorithm realizations for each set of parameter values. One can see that there is a narrow region in which the objective function reaches its minimum on the grid cells where $P_e = 0.035$. Next, we examine more closely the dependence of the objective function distribution on parameter $P_o$ for fixed $P_e = 0.035$, which is presented in Figure 10B. One can see that for a long range of parameter $P_o$, the objective function values remain low, until they increase for large values $P_o > 0.8$, where the maximum degree of graphs become too large (the dependence of topological metrics on $P_o$ is presented in Figure S1). From Figure 10, one can estimate the following ranges of viable parameter values for $P_e$ and $N_l$: $P_e \in (0.001, 0.07)$, $P_o \in (0.01, 0.8)$. Two examples of rule-based graphs with satisfactory topological fitness generated by the algorithm with the use of optimal parameters $P_e = 0.035$, $P_o = 0.21$ are shown in Figure 7.

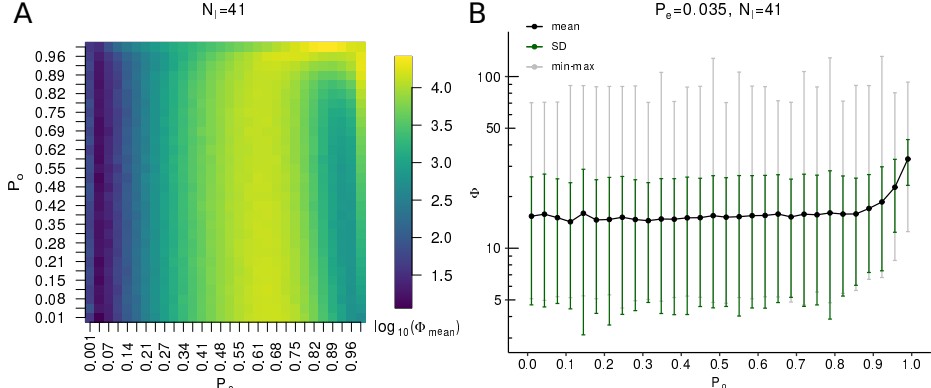

**Figure 10.** The landscape of topological fitness of rule-based graphs over different values of algorithm parameters. (**A**) Image of $\log_{10}(\Phi_{mean})$ values as function of varying parameters $P_e$ and $P_o$ and fixed $N_l = 41$. The mean values are calculated in each image cell over 640 graphs realizations of rule-based graphs generated with corresponding parameter sets. (**B**) Dependence of objective function on $P_o$ with fixed $P_e = 0.035$, $N_l = 41$. The statistics on values of objective function (14) are calculated over 10,000 realizations of the algorithm for each value of $P_o$.

## 5. Comparative Analysis of the HLS Graph Models

The differences between the anatomy data-based and the rule-based graphs in terms of the metrics introduced in Section 4 are summarized in Tables 1 and 2. The histograms of robustness distributions ($R_k$ values) for each type of graph calculated over $N_r = 100,000$ ($N_r = 435$ for Reddy's graph is the number of random deletions of 5% of nodes corresponding to 29 one node deletions and 406 deletions of randomly chosen pairs of nodes) realizations of random deletion of 5% of vertices are presented in Figure 11. The distributions of flow diversity of anatomy-based graphs are presented in Figure 12. Please note that for data-based graph we use the flows computed in Section 2.2.2. For Reddy's graph we obtain the flows assuming the equidistant inflow along 16 input edges equal to the 1/16-th of the outflow to the blue output node and the conservation law.

**Table 1.** Summary statistics for Reddy's, anatomy-based and rule-based graphs characterizing their topological properties. For algorithmically generated graphs, we present the statistics obtained over 10,000 graphs generated with algorithm parameters $P_e = 0.035$, $P_o = 0.21$, $N_l = 41$. Robustness was calculated over 1000 algorithm realizations, with 1000 removal attempts for each graph.

| Lymphatic Vascular System Graph Model | Reddy's Model | Anatomy-Based Model | Rule-Based Model (Mean) | Rule-Based Model (SD) | Rule-Based m. (Min-Max Range) |
|---|---|---|---|---|---|
| $G(n, m)$ | $(29, 28)$ | $(996, 1117)$ | $(996, 1029)$ | $(0, 6.4)$ | $(996, 1009–1056)$ |
| $N_{inp}$ | 16 | 357 | 357 | 0 | $(357–357)$ |
| Maximum degree, $\Delta_G$ | 4 | 8 | 16 | 1.58 | $(8–21)$ |
| Girth, $g$ | 0 | 3 | 4 | 0.9 | $(3–14)$ |
| Diameter, $D$ | 5 | 40 | 39.96 | 0.22 | $(37–40)$ |
| Radius, $r$ | 4 | 30 | 28 | 2.3 | $(23–38)$ |
| Average path length, $l_G$ | 2.46 | 12.79 | 15.3 | 0.86 | $(13.6–18)$ |
| Energy, $En$ | 32.1 | 1224.5 | 1190 | 4.9 | $(1173–1203)$ |
| Spectral radius, $\rho$ | 2.58 | 3.51 | 4.18 | 0.19 | $(3.28–4.72)$ |
| Edge density, $\rho_d$ | 0.034 | 0.001127 | 0.001038 | $6.4 \cdot 10^{-6}$ | $(0.00102–0.00107)$ |
| Clustering coefficient, $C$ | 0 | 0.027 | 0.0004 | 0.0008 | $(0–0.0036)$ |
| Number of separators, $n_{sep}$ | 13 | 401 | 496 | 20 | $(437–548)$ |
| Robustness, $R$ | 0.7 | 0.6 | 0.66 | 0.05 | $(0.45–0.77)$ |
| Average flow diversity, $\overline{D_{flow}}$ | 0.908 | 0.996 | – | – | – |

**Table 2.** Summary statistics for the subgraphs of the anatomy-based graph which correspond to different parts of the body.

| Subgraph | Left Arm | Right Arm | Head & Neck | Torso | Left Leg | Right Leg |
|---|---|---|---|---|---|---|
| $G(n, m)$ | $(149, 181)$ | $(141, 170)$ | $(198, 208)$ | $(168, 183)$ | $(163, 176)$ | $(177, 192)$ |
| Number of input nodes | 46 | 45 | 64 | 68 | 66 | 71 |
| Number of output nodes | 1 | 1 | 2 | 1 | 1 | 1 |
| Maximum degree, $\Delta_G$ | 8 | 8 | 4 | 8 | 5 | 4 |
| Girth, $g$ | 3 | 3 | 4 | 3 | 3 | 3 |
| Diameter, $D$ | 23 | 22 | 21 | 18 | 22 | 22 |
| Radius, $r$ | 10 | 9 | 13 | 9 | 13 | 13 |
| Average path length, $l_G$ | 8.77 | 8.37 | 7.24 | 7.02 | 8.95 | 9.55 |
| Energy, $En$ | 192.22 | 180.6 | 236.24 | 164.4 | 196.57 | 214.1 |
| Spectral radius, $\rho$ | 3.3 | 3.32 | 2.95 | 3.51 | 2.92 | 3.08 |
| Edge density, $\rho_d$ | 0.0082 | 0.0086 | 0.0053 | 0.0065 | 0.0067 | 0.0062 |
| Clustering coefficient, $C$ | 0.049 | 0.035 | 0 | 0.055 | 0.01 | 0.009 |
| Number of separators, $n_{sep}$ | 41 | 41 | 46 (left), 46 (right) | 81 | 66 | 72 |
| Number of LNs | 36 | 36 | 51 (left), 51 (right) | 59 | 21 | 18 |

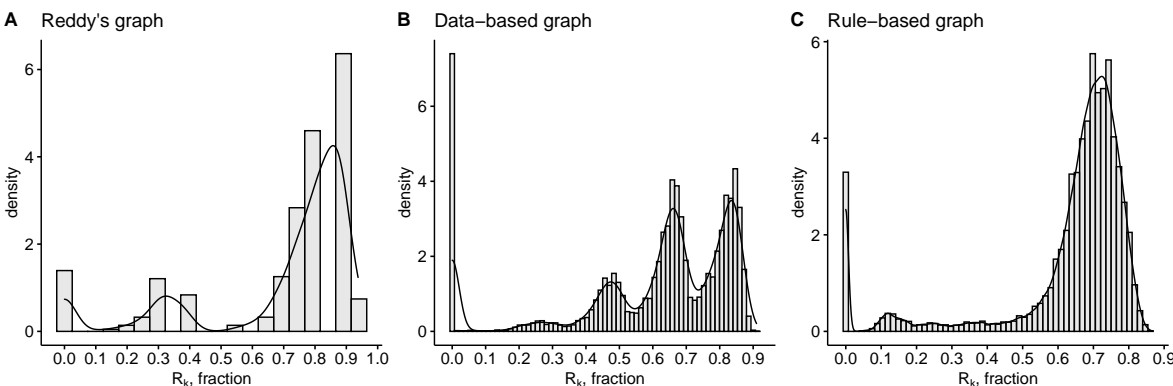

**Figure 11.** Histograms of the robustness of (**A**) Reddy's graph, (**B**) anatomical data-based graph and (**C**) rule-based random graph presented in Figure 7A.

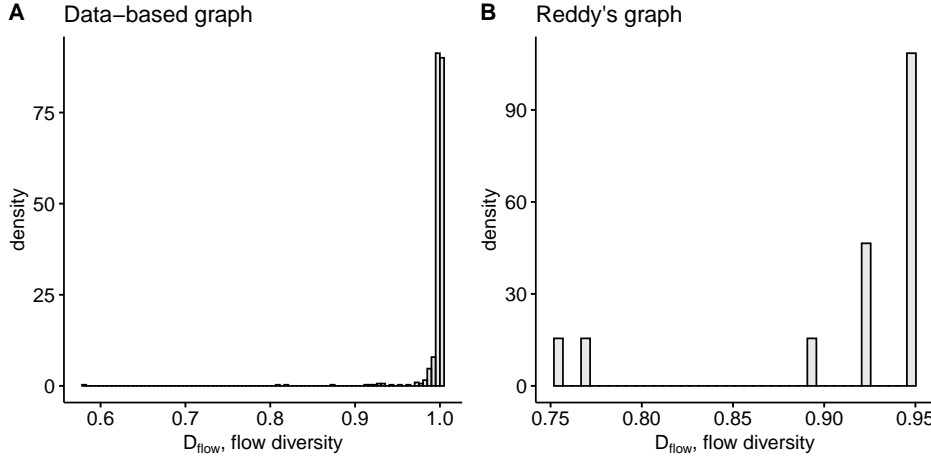

**Figure 12.** Histograms of the flow diversity of the inner vertices of anatomy-based graphs: (**A**) for data-based graph, (**B**) for Reddy's graph.

Given the algorithmically generated rule-based graph with optimal parameters presented in Figure 7A, we used another method to estimate the level of its resemblance to the anatomy-based graph. Specifically, to evaluate the level of isomorphism ($LI$) between the data-based and the rule-based graphs we searched for the row and column permutations to find the best match of them. Please note that examination of the graph isomorphism level belongs to the NP complexity calls. For this, we formed an expanded connectivity matrix $M_1 = \begin{pmatrix} A & O \\ O & O \end{pmatrix}$, $M_2 = \begin{pmatrix} O & O \\ O & B \end{pmatrix}$, $D_{max} = \sum_{i,j}(M_{1_{i,j}} + M_{2_{i,j}})$, where $A$ is the adjacency matrix of the data-based graph, and $B$ is the adjacency matrix of the rule-based graph, $O$ is the zero matrix with equal sizes, $D_{max}$ is the initial Hamming distance for the given adjacency matrices. We used the simulated annealing method to minimize the Hamming distance between matrices $M_1$, $M_2$ by performing permutations in the $M_2$ matrix (based on graph isomorphism):

$$\sum_{i,j} |M_{1_{i,j}} - M_{2_{i,j}}| \rightarrow min \tag{15}$$

The permutations in the $M_2$ matrix are performed as follows: a pair of numbers $i$, $j$ is chosen at random. Then, in the matrix $M_2$, columns number $i$, $j$ are swapped. After that, the same operation is performed with rows $i$, $j$. From the point of view of the graph, such transformations mean a permutation of the designations of the vertices $i$, $j$. Using this, we carry out a stochastic optimization of the $M_2$ matrix by the simulated annealing method to minimize (15). After that, we calculate the relative distance between graphs, presented by the adjacency matrices $M_1$, $M_2$ as $D_{rel} = \frac{\sum_{i,j} |M_{1_{i,j}} - M_{2_{i,j}}|}{D_{max}}$. Here, $D_{rel} = 1$ means a complete mismatch, $D_{rel} = 0$ means the complete equivalence of the adjacency matrices. By solving the minimization problem for the entire graphs, we achieved the upper bound estimate of the mismatch level of the range $LI \leq 0.49$.

## 6. Conclusions

The human lymphatic system is a complex system consisting of many components and performing several important functions related to the metabolism, fluid tissue homeostasis and the immune system [21,22]. Currently, there are only a few models addressing the network structure of the human lymphatic system in a systematic way [3–5]. In this work, we propose an approach based on the graph theory to modeling the human lymphatic system and analyzing the fundamental mathematical properties of the resulting graph models. Two different types of graph models are developed, the anatomical data- and the rule-based ones. The transformation of the anatomical data-based simple graph into the oriented graph is implemented on the base of the steady-state lymph flow balance in the

lymphatic vessel network. This homeostatic balance is quantified by applying the Poiseuille equation to every vessel and the mass conservation law to every vessel junction. The available empirical data on lymph flow through the system are rather scarce and range from 0.004 mm$^3$/s in rats to 46.3 mm$^3$/s in humans [2]. Our computations consistently suggest the range of lymph flows from 0.001 to 10 mm$^3$/s, depending on the location of the respective vessel.

Our study is aimed to understand the fundamental topological properties of the human lymphatic system and to critically analyze the underlying organizational principles (rules) of the LS that have been proposed for modeling the systemic spreading of HIV infection. The combination of the developed HLS data-based graph and the publicly available software SimVascular (https://simvascular.github. io/index.html) designed for blood flow simulations provide a solid basis to further move on to the 1D, 2D- and 3D-simulations of the lymphatic system flows. The knowledge gained on the lymphatic network topology can be useful for systems pharmacokinetics studies and drug therapies design.

Recent progress in tissue bioengineering and 3D bio-printing enables the development of artificial organs including those of the lymphatics system, e.g., the LN-on-a-chip [23]. The artificial LNs can be used as a functional cure for certain pathological conditions like lymphoedema characterized by disruption of the interstitial fluid transport. The graph model of the human lymphatic system provides a platform for an optimal positioning of the artificial LNs and lymphatic vessels to restore the functioning and performance of the HLS.

The lymphatic system (LS) is responsible for maintaining the fluid balance in tissues and the functioning of the immune system. As the complexity of the organization and regulation of the immune system is extremely high, the move towards a mechanistic understanding of the immune system functioning takes the form of specific rules (see [24–27]). The respective set of organizational principles (rules) of the LS structure was recently proposed and used for modeling the systemic spreading of HIV infection [9]. In our study we analyzed the consistency of the rule-based graph of the human lymphatic system with that based on the anatomical data. To this end, a novel computational algorithm for generation of the rule-based random graph has been developed. Some fundamental characteristics of the two types of HLS graph models are analyzed using metrics such as graph energy, clustering, robustness, etc.

Additionally, we estimated the optimal parameters of the proposed algorithm for generating the rule-based graph model which provide the best-fit to the anatomy data-based graph model with respect to basic topological metrics. Our analysis revealed that the currently available set of rules specifying universal properties of HLS for the rule–based graph modeling requires further maturation to be applied in clinical studies.

Overall, it is the first study in which the full complexity of the HLS network is addressed by methods of the graph theory methods. We probe to elucidate the efficacy of the rule-based approach to generate the HLS models. Further specification of the basic rules underlying the structure and function of the HLS is needed in collaboration with immunologists and physiologists. The graph models which we make publicly available, provide an important step towards a quantitative analysis of transport phenomena in the whole lymphatic system. This is a problem of a paramount importance for systems immunology, pharmacokinetics and medicine [1,2,28].

**Supplementary Materials:** The following are available online at http://www.mdpi.com/2227-7390/8/12/2236/s1, Figure S1: extended rule-based graphs metrics analysis; graph_edges.txt, graph_vertices.txt: data-based graph of the HLS; graph_generator.cpp, Makefile: C++ code to build rule-based graphs.

**Author Contributions:** Conceptualization, G.B., R.S. and D.G.; methodology, R.S., D.G., I.S., V.C. and G.B.; software, R.S., D.G., D.P.; validation, R.S., D.G. and I.S.; formal analysis, R.S., D.G.; investigation, R.S., D.G., D.P.; data curation, R.S., D.G.; writing—original draft preparation, R.S., D.G., G.B.; writing—review and editing, R.S., D.G., G.B., V.C., I.S.; supervision, G.B.; project administration, G.B.; funding acquisition, G.B. All authors have read and agreed to the published version of the manuscript.

**Funding:** This research was funded by the Russian Science Foundation grant number 18-11-00171. R.S., D.G. and G.B. were partly supported by Moscow Center for Fundamental and Applied Mathematics (agreement with the Ministry of Education and Science of the Russian Federation No. 075-15-2019-1624) and by RFBR (project number 20-01-00352). R.S. Savinkov was partly supported by the «RUDN University Program 5-100».

**Conflicts of Interest:** The authors declare no conflict of interest. The funders had no role in the design of the study; in the collection, analyses, or interpretation of data; in the writing of the manuscript, or in the decision to publish the results.

## Abbreviations

The following abbreviations are used in this manuscript:

HLS   Human Lymphatic System
LNs   lymph nodes

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
