# Peer review of "Graph Theory for Modeling and Analysis of the Human Lymphatic System"

_mathematics, doi:10.3390/math8122236_

Round 1

Reviewer 1 Report

The authors have done all that was asked of them, and the quality of the paper, its results and discussion have significantly improved after using the anatomical data to guide the rules of the rule-based graph.

Author Response

We thank the Reviewer for the thorough work on our manuscript and positive evaluation of our study.

Reviewer 2 Report

The manuscript has been thoroughly revised to address the comments and warrants publication. 

Author Response

(The authors gave the same response as above.)

Reviewer 3 Report

The corrections made allow us to better understand the 'spirit' of the paper and its usefulness in the practice. Perhaps it would have been better (but this is already enough!) to make a further clarification on the clinical 'applicability' of the results, especially in the case of impared development of part of the lymphatic system as occurs, for example, in primary lymphoedema.

Author Response

We thank the Reviewer for the thorough work on our manuscript and positive evaluation of our study. In response to your suggestion, the following comment to the Discussion section has been added:

"Recent progress in tissue bioengineering and 3D bio-printing enables the development of artificial organs including those of the lymphatics system, e.g. the LN-on-a-chip [29]. The artificial LNs can be used as a functional cure for certain pathological conditions like lymphoedema characterized by disruption of the interstitial fluid transport. The graph model of the human lymphatic system provides a platform for an optimal positioning of the artificial LNs and lymphatic vessels in order to restore the functioning and performance of the HLS."

This manuscript is a resubmission of an earlier submission. The following is a list of the peer review reports and author responses from that submission.

Round 1

Reviewer 1 Report

The topic treated is interesting and the theoretical demonstrations are well articulated and documented. However, it does not seem likely to be able to standardize the study presented in the human living, above all for two fundamental reasons:

- the composition of the lymph varies daily as a whole and in every single organ, both from a qualitative and quantitative point of view. This means that the same viscosity, which affects the lymphoreology, varies and alters the proposed mathematical formulas.

  • the lymphatic system in the human species has a development that is comparable to the 'gray scale': it goes from white to black. This explains, for example, why 25% of patients undergoing axillary lymphadenectomy undergo secondary lymphedema (with variable time intervals from the operation) while 75% remain along the life with coincident limbs. It is practically impossible to adapt a mathematical model to such a high and non-standardized variability; and this both from the point of view of the volumes and the flow of fluids in the unit of time.
  • Interesting work but lacking, in my opinion, of a practical / clinical usefulness.

Reviewer 2 Report

The authors present a graph-theory-based approach to model and analyze network models of the human lymphatic system (HLS). Two different types of graphs are developed, the anatomical-data-based (ADB) and the rule-based (RB). ADB is implemented by quantifying the steady state fluid balance using the Poiseuille equation and mass conservation. A computational algorithm for the generation of the RB random graph is developed. Fundamental characteristics of the two types of HLS graph models are analyzed using metrics such as graph energy.

The following points need to be addressed for reconsideration of the manuscript for publication.

  1. One of the hallmarks of science is reproducibility. How do the authors expect the readers to verify, expand, or use their results if the underlying data is not freely available? The authors comment on the length of the lymphatic vessels. Where can one find the coordinates, or the splines that generate these structures?
  2. Algorithm 1 creates a random graph, and the authors call it (see Figure 10) a rule-based generated directed random graph of the HLS. How is the lymph node anatomy, e.g., position in space, used to qualify it as a human lymphatic system?
  3. What are the advantages in clinic of using the rule-based graph?
  4. The use of ternary operators in equation 8 (x ? Y : z) is not recommended since it is not a universally accepted notation.
  5. It is not clear how the degree of isomorphism between the graphs was computed. This has to be thoroughly documented or supported with enough references. This point is important since the authors have to defend the position that the random graph method has some anatomical or mathematical significance.
  6. The authors state: The graph models which we make publicly available, provide an important step towards a quantitative analysis of transport phenomena in the whole lymphatic system. How are they making the data publicly available? How can researchers use their lymphatic system model to study the kinetics of immune cells or infection or drugs without the raw data? Why not use GitHub to make public the data sets and the code used to process the data?
  7. How does this tool compare to other truly publicly available software that can solve Poiseuille's flow (among others) like SimVascular?
  8. Why not compare the computed flow rates with physiologically accurate values to validate the model?
  9. In Figure 10, there is not red dot as stated in the caption. Maybe the authors are referring to the blue dot?

Reviewer 3 Report

General comments The authors analyse two data-based graph models of the human lymphatic system and compare them to a rule-based random graph based on a very general set of rules. One of the data-based graph models is a directed graph with a smaller number of nodes (29), while the other (Plastic boy) model is an undirected graph with a larger number of nodes (841). They transform the undirected data-based graph into a directed graph using Poiseuille law and mass conservation law. The statistical analysis of the two data-based graphs gives consistent results. However, the random graph with 841 nodes created using their algorithm does not match the data-based graphs at all. It seems to represent a negative result: their algorithm is not suitable for the generation of a graph model of the lymphatic system.   None of the data-based graph statistics were used in the creation of the rule-based graph. While the Plastic boy model's transformation into a directed graph using hydrodynamics is interesting and rather ingenious, this approach was not used to improve the rule-based graph. One reason is probably that the rule-based graph does not have the lengths of the edges needed for Eqs.1 and 3. This could have been perhaps remedied by assigning lengths from the data-based length distribution. All this makes one wonder, why analyse two data-based graphs just to create one completely different graph? I agree that understanding the lymphatic system is of great importance, but I do not see how a graph that does not represent it contributes to this understanding.   In their conclusion, the authors state that they plan to improve their rule-based model in the future. I suggest that for now, they take out the section about the rule-based graph, just keeping the Plastic boy graph's transformation to an oriented graph.   Specific comment Figure 6 has neither axis labels nor any explanation of the colour (heat) bar on the right side.